## [Decision Letter · Decision Letter 0]

26 Aug 2025

Dear Dr. DRAOU,

We look forward to receiving your revised manuscript.

Kind regards,

Yogesh Kumar

Academic Editor

PLOS ONE

2. Please note that your Data Availability Statement is currently missing [the repository name and/or the DOI/accession number of each dataset OR a direct link to access each database]. If your manuscript is accepted for publication, you will be asked to provide these details on a very short timeline. We therefore suggest that you provide this information now, though we will not hold up the peer review process if you are unable.

Additional Editor Comments (if provided):

Reviewers' comments:

Reviewer's Responses to Questions

**Comments to the Author**

1. Is the manuscript technically sound, and do the data support the conclusions?

Reviewer #1: No

Reviewer #2: Yes

2. Has the statistical analysis been performed appropriately and rigorously?

Reviewer #1: I Don't Know

Reviewer #2: Yes

3. Have the authors made all data underlying the findings in their manuscript fully available?

Reviewer #1: No

Reviewer #2: Yes

4. Is the manuscript presented in an intelligible fashion and written in standard English?

Reviewer #1: No

Reviewer #2: Yes

Reviewer #1: Dear Author,

your manuscript ‘Structural 1 characterization and antifungal properties of sequentially extracted polysaccharides from Algerian Opuntia ficus-indica L. cladodes’ is not suitable, in my opinion, for publication. It presents too many flaws that are summarized in the attached file (for the first part of the manuscript). Please, be aware of the advices and improve the manuscript before submitting to a more specialized journal.

Reviewer #2: The whole manuscript has been planned and executed well. I would suggest a minor revision considering the following observations:

1. The introduction needs to include the importance of extracting polysaccharides from such kind of plant material/ substrates, highlighting the economics.

2. The conclusion needs to be crisp with the findings and should include the potential applications of extracted polysaccharides.

3. Since it is a new method for simultaneous extraction of different polysaccharide fractions as per the claim, the benefit of this method over conventional methods should be highlighted in the introduction as well as the conclusion part.

**Do you want your identity to be public for this peer review?** For information about this choice, including consent withdrawal, please see our Privacy Policy

Reviewer #1: No

Reviewer #2: No

---

## [Author Response · Author response to Decision Letter 1]

24 Sep 2025

Re: Revision of Manuscript PONE-D-25-37900

Dear PLOS ONE Editorial Team,

Thank you for the opportunity to revise our manuscript. We are grateful to the reviewers and editors for their time and insightful comments, which have significantly improved the quality of our work. We have carefully addressed all points raised in the decision letter.

Below is our point-by-point response to the specific comments:

Response to Academic Editor & Reviewer #1:

We thank the reviewers for their thorough critique. We have completely restructured the manuscript to address the concerns regarding technical soundness, methodology description, and language clarity. Key changes include:

Comment (Nota): "Here you can add details of work" (re: duplicated sentence).

Response: We have removed the duplication and expanded the narrative to better frame the knowledge gap and our study's contribution (Page 3, Lines 57-66).

Comment (Nota): "M&M section is not well written: only methodology must be reported..."

Response: We have entirely rewritten the Materials and Methods section (Sections 2.1 & 2.2) to be purely descriptive, removing all justifications and consistently using the past passive tense (Pages 4-6, Lines 75-120).

Comment: "Describe the stage of development, the season, and the amount of sampling."

Response: We have added explicit details: *"Cladodes were harvested in November from approximately 10-year-old plants... A single cladode per plant was used"* (Page 4, Lines 76-78).

Comment: "for how long" (lyophilization).

Response: We have added the specific duration: "lyophilized for 5 days" (Page 4, Line 83).

Comment: "In M&M at least a brief description is presented... Moreover, the figure are not in good quality..."

Response:

Methodology: We replaced the brief mention with a detailed, step-by-step protocol (Pages 5-6, Lines 86-120).

Figures: We confirm high-resolution TIF files were initially submitted. As recommended, we have now processed Fig. 1 and Fig. 2 through the PACE tool and uploaded the optimized files to ensure they meet all technical standards.

Comment (Nota): "This is not M&M, it is a lab protocol!"

Response: We have reformulated the section to remove imperative verbs and use descriptive past passive tense (Page 9, Lines 181-188).

Comment: "In fact you did not reported!" (standardization).

Response: We have replaced the vague phrase with explicit sampling details (Page 9, Lines 189-191).

Comment: "Are the references as for the 'Guide for Authors' format of the journal?"

Response: We have comprehensively reformatted the entire bibliography to strict PLOS ONE style, including author initials, DOI additions, and correct volume/page formatting (Pages 31-36).

Response to Reviewer #2:

We thank the reviewer for their positive assessment and valuable suggestions to enhance the impact of our work.

Comment: "The introduction needs to include the importance... highlighting the economics."

Response: We have added a sentence on the economic potential and valorization of native species (Page 3, Lines 53-55).

Comment: "The conclusion needs to be crisp... and should include the potential applications."

Response: We have revised the conclusion to be more concise and explicitly state the applications as sustainable preservatives (Page 22, Lines 493-496).

Comment: "highlight the benefit of this method over conventional methods"

Response: We have emphasized the novelty and advantage of our sequential protocol in both the introduction (Page 3, Lines 64-66) and conclusion (Page 24, Lines 506-510).

All data underlying the findings are fully available without restriction on Figshare (DOI: 10.6084/m9.figshare.29508983.v4).

We believe our revisions have thoroughly addressed all concerns and that the manuscript now meets PLOS ONE's publication criteria. Thank you again for your guidance.

Sincerely,

Nassima DRAOU, Ph.D.

On behalf of all co-authors.

---

## [Decision Letter · Decision Letter 1]

8 Jan 2026

We look forward to receiving your revised manuscript.

Kind regards,

Nishant Kumar, Ph.D

Academic Editor

PLOS One

Journal Requirements:

Reviewers' comments:

Reviewer's Responses to Questions

**Comments to the Author**

Reviewer #1: (No Response)

Reviewer #3: All comments have been addressed

2. Is the manuscript technically sound, and do the data support the conclusions?

Reviewer #1: Yes

Reviewer #3: Yes

3. Has the statistical analysis been performed appropriately and rigorously?

Reviewer #1: Yes

Reviewer #3: Yes

4. Have the authors made all data underlying the findings in their manuscript fully available?

Reviewer #1: Yes

Reviewer #3: Yes

5. Is the manuscript presented in an intelligible fashion and written in standard English?

Reviewer #1: Yes

Reviewer #3: Yes

Reviewer #1: Dear Authors,

your revised manuscript ‘Structural characterization and antifungal properties of sequentially extracted polysaccharides from Algerian Opuntia ficus-indica L. cladodes’ has been improved. However, additional amendments are required to make it acceptable.

Please, see the comments on the attached file.

Reviewer #3: The authors appear to have made the necessary corrections. The work can be accepted for publication in its current form.

**Do you want your identity to be public for this peer review?** For information about this choice, including consent withdrawal, please see our Privacy Policy

Reviewer #1: No

Reviewer #3: **Yes:** Murat Yılmaz

---

## [Author Response · Author response to Decision Letter 2]

19 Jan 2026

Dear Dr. Nishant Kumar and Reviewers,

Thank you for the opportunity to revise our manuscript and for your constructive comments, which have significantly improved the clarity and scientific rigor of our work. We have addressed all points raised, as detailed below. All changes are highlighted in the revised manuscript with tracked changes, and a clean version is also provided.

Reviewer #1

Comment 1: “Corresponding Author section – remove ‘Field Code Changed’ and formatting artifacts.”

Response: All field codes and formatting artifacts (e.g., “Field Code Changed”, “Formatted: English…”) have been removed from the manuscript.

Comment 2: “Introduction – duplicated sentences (lines 65–68) and redundant phrasing (line 78).”

Response: The duplicated sentences and redundant phrase have been deleted. The introduction is now concise and flows logically.

Comment 3: “Materials & Methods – unclear ethanol precipitation ratio (line 112).”

Response: We have clarified the ratio: “precipitated with absolute ethanol at a 1:3 (v/v) ratio (one volume of aqueous extract to three volumes of ethanol).”

Comment 4: “Materials & Methods – unclear description of the three extraction steps (lines 115, 120).”

Response: We have added an explanatory sentence: “Each extraction was performed on the residual solid from the previous step to sequentially recover water-soluble polysaccharides.”

Comment 5: “Materials & Methods – duplicated paragraphs in section 2.6 (lines 202–209).”

Response: The redundant description of the antifungal assay has been removed, retaining only the clear, single protocol.

Comment 6: “Statistical information – move sentence on biological replicates to statistical section (line 211).”

Response: The sentence “Three independent biological replicates (n=3) were analyzed for each experiment.” has been moved to Section 2.7 (Statistical Analysis).

Comment 7: “Results – remove irrelevant reference in Water Content section (lines 247–248).”

Response: The reference to “Dittrichia viscosa (Bouri et al., 2021)” has been removed.

Comment 8: “Results – Section 3.5 should be rewritten as a narrative, not in bullet points.”

Response: Section 3.5 has been completely rewritten into a continuous, discursive narrative that guides the reader through the findings.

Comment 9: “Discussion – remove ‘for Algerian cell walls’ (line 517).”

Response: The phrase has been deleted.

Comment 10: “Conclusion – do not start with a number; rewrite to interpret results, not summarize.”

Response: The conclusion has been rewritten to begin with a proper sentence, interpret the main findings in light of the research questions, and include limitations and future directions.

Comment 11: “Bibliography – clean formatting artifacts.”

Response: All formatting marks in the reference list have been removed, and the bibliography now conforms to PLOS ONE style.

Reviewer #3

Comment: “All comments have been addressed.”

Response: We thank the reviewer for their positive evaluation.

Additional Technical Corrections

• All figures and tables are correctly cited.

• Data availability statement is complete and accessible via Figshare.

• The manuscript meets PLOS ONE submission requirements.

We believe the revised manuscript now fully addresses all reviewers’ concerns and is suitable for publication in PLOS ONE. Thank you again for your valuable time and guidance.

Sincerely,

Nassima DRAOU, Ph.D.

---

## [Decision Letter · Decision Letter 2]

3 Feb 2026

Dear Dr. DRAOU,

Thank you for submitting your manuscript to PLOS ONE. After careful consideration, we feel that it has merit but does not fully meet PLOS ONE’s publication criteria as it currently stands. Therefore, we invite you to submit a revised version of the manuscript that addresses the points raised during the review process.

We look forward to receiving your revised manuscript.

Kind regards,

Nishant Kumar, Ph.D

Academic Editor

PLOS One

Journal Requirements:

Reviewers' comments:

Reviewer's Responses to Questions

**Comments to the Author**

Reviewer #1: All comments have been addressed

Reviewer #3: All comments have been addressed

2. Is the manuscript technically sound, and do the data support the conclusions?

Reviewer #1: Yes

Reviewer #3: Yes

3. Has the statistical analysis been performed appropriately and rigorously?

Reviewer #1: Yes

Reviewer #3: Yes

4. Have the authors made all data underlying the findings in their manuscript fully available?

Reviewer #1: Yes

Reviewer #3: Yes

5. Is the manuscript presented in an intelligible fashion and written in standard English?

Reviewer #1: Yes

Reviewer #3: Yes

Reviewer #1: Dear Authors,

your revised manuscript ‘Structural characterization and antifungal properties of sequentially extracted polysaccharides from Algerian Opuntia ficus-indica L. cladodes’ has been improved. However, few changes are needed to make it acceptable.

In Table 4: change, at line 361, ’glucuronic’ non ‘glucoronic’

The same, change ‘galA’, not ‘Gala’

Line 491 when ref Liu et al 2021: become ref 6. The same for ref Zhang et al 2023, become ref 5.

In Results and Discussion, when dealing with the antifungal activity of polysaccharide, specify ‘pectic and hemicellulose polysaccharides’ as you only tested those polysaccharides. You did not tested mucilage polysaccharides for es. Therefore, specify this point in every place it is needed.

Reviewer #3: The authors appear to have made the necessary corrections. The work can be accepted for publication in its current form.

**Do you want your identity to be public for this peer review?** For information about this choice, including consent withdrawal, please see our Privacy Policy

Reviewer #1: No

Reviewer #3: No

---

## [Author Response · Author response to Decision Letter 3]

6 Feb 2026

Dear Dr. Nishant Kumar and Reviewers,

Thank you for your positive evaluation and for the opportunity to submit a revised version of our manuscript. We are grateful for the constructive comments, which have helped us further improve the clarity and precision of the work.

Below, we provide a point-by-point response to the remaining comments.

Response to Reviewer #1:

We sincerely thank Reviewer #1 for their careful reading and helpful suggestions. All points have been addressed as follows:

1. Table 4, line 356: “glucoronic” has been corrected to “glucuronic,” and “Gala” has been changed to “GalA” to accurately reflect standard abbreviations for galacturonic acid.

2. Citations in text (lines 469–470): The references to Liu et al., 2021 and Zhang et al., 2023 have been updated to the correct citation numbers [6] and [5], respectively.

3. Clarification of tested polysaccharides: Throughout the Results and Discussion sections, we have specified that the antifungal activity refers specifically to “pectic and hemicellulose polysaccharides” (e.g., lines 418, 494), making it clear that mucilage polysaccharides were not included in the bioactivity assays.

Response to Reviewer #3:

We thank Reviewer #3 for their supportive feedback and confirmation that the manuscript is acceptable in its current form.

All changes are highlighted in the attached “Revised Manuscript with Track Changes”. A clean version is also provided.

We believe the manuscript now fully addresses all reviewers’ concerns and meets the standards of PLOS ONE.

Thank you again for your time and consideration.

Sincerely,

Nassima DRAOU, Ph.D.

---

## [Decision Letter · Decision Letter 3]

20 Feb 2026

Structural characterization and antifungal properties of sequentially extracted polysaccharides from Algerian Opuntia ficus-indica L. cladodes.

PONE-D-25-37900R3

Dear Dr. Draou,

We’re pleased to inform you that your manuscript has been judged scientifically suitable for publication and will be formally accepted for publication once it meets all outstanding technical requirements.

Kind regards,

Nishant Kumar, Ph.D

Academic Editor

PLOS One

Additional Editor Comments (optional):

Reviewers' comments:

Reviewer's Responses to Questions

**Comments to the Author**

Reviewer #1: All comments have been addressed

2. Is the manuscript technically sound, and do the data support the conclusions?

Reviewer #1: Yes

3. Has the statistical analysis been performed appropriately and rigorously?

Reviewer #1: Yes

4. Have the authors made all data underlying the findings in their manuscript fully available?

Reviewer #1: Yes

5. Is the manuscript presented in an intelligible fashion and written in standard English?

Reviewer #1: Yes

Reviewer #1: Dear Authors,

your manuscript have been improved.

My idea is the manuscript is suitable for publication in the present form.

**Do you want your identity to be public for this peer review?** For information about this choice, including consent withdrawal, please see our Privacy Policy

Reviewer #1: No

---

## [Editor Report · Acceptance letter]

PONE-D-25-37900R3

PLOS One

Dear Dr. DRAOU,

I'm pleased to inform you that your manuscript has been deemed suitable for publication in PLOS One. Congratulations! Your manuscript is now being handed over to our production team.

Kind regards,

on behalf of

Dr. Nishant Kumar

Academic Editor

PLOS One